# Acute Improvements of Oxygenation with Cpap and Clinical Outcomes in Severe COVID-19 Pneumonia: A Multicenter, Retrospective Study

**DOI:** 10.3390/jcm11237186

**Published:** 2022-12-02

**Authors:** Stefano Pini, Dejan Radovanovic, Marina Saad, Marina Gatti, Fiammetta Danzo, Michele Mondoni, Stefano Aliberti, Stefano Centanni, Francesco Blasi, Davide Alberto Chiumello, Pierachille Santus

**Affiliations:** 1Division of Respiratory Diseases, ASST Fatebenefratelli-Sacco, Ospedale Luigi Sacco, Polo Universitario, 20157 Milano, Italy; 2Department of Biomedical and Clinical Sciences (DIBIC), Università Degli Studi Di Milano, 20157 Milano, Italy; 3Respiratory Unit, ASST Santi Paolo e Carlo, San Paolo Hospital, 20142 Milano, Italy; 4Dipartimento di Scienze Della Salute, Università Degli Studi Di Milano, 20146 Milano, Italy; 5Department of Biomedical Sciences, Humanitas University, 20090 Pieve Emanuele, Italy; 6Respiratory Unit, IRCCS Humanitas Research Hospital, 20089 Rozzano, Italy; 7Respiratory Unit and Cystic Fibrosis Adult Center, Fondazione IRCCS Ca’ Granda Ospedale Maggiore Policlinico, 20122 Milano, Italy; 8Department of Pathophysiology and Transplantation, Università Degli Studi Di Milano, 20122 Milano, Italy; 9Department of Anesthesia and Intensive Care, ASST Santi Paolo e Carlo, San Paolo University Hospital, 20142 Milano, Italy; 10Coordinated Research Center on Respiratory Failure, Università Degli Studi Di Milano, 20142 Milano, Italy

**Keywords:** continuous positive airway pressure, COVID-19, acute respiratory failure, pneumonia, blood oxygen levels, mortality

## Abstract

It is not known if the degrees of improvement in oxygenation obtained by CPAP can predict clinical outcomes in patients with COVID-19 pneumonia. This was a retrospective study conducted on patients with severe COVID-19 pneumonia treated with CPAP in three University hospitals in Milan, Italy, from March 2020 to March 2021. Arterial gas analysis was obtained before and 1 h after starting CPAP. CPAP failure included either death in the respiratory units while on CPAP or the need for intubation. Two hundred and eleven patients (mean age 64 years, 74% males) were included. Baseline median PaO_2_, PaO_2_/FiO_2_ ratio (P/F), and the alveolar-arterial (A-a) O_2_ gradient were 68 (57–83) mmHg, 129 (91–179) mmHg and 310 (177–559) mmHg, respectively. Forty-two (19.9%) patients died in the respiratory units and 51 (24.2%) were intubated. After starting CPAP, PaO_2_/FiO_2_ increased by 57 (12–113; *p* < 0.001) mmHg, and (A-a) O_2_ was reduced by 68 (−25–250; *p* < 0.001) mmHg. A substantial overlap of PaO_2_, P/F, and A-a gradient at baseline and during CPAP was observed in CPAP failures and successes; CPAP-associated improvements in oxygenation in both groups were similar. In conclusion, CPAP-associated improvements in oxygenation do not predict clinical outcomes in patients with severe COVID-19 pneumonia.

## 1. Introduction

Continuous positive airway pressure (CPAP) has been widely employed for respiratory support in patients with acute hypoxic respiratory failure (hARF) due to pneumonia associated with coronavirus disease 2019 (COVID-19) [1]. Application of positive end expiratory pressure (PEEP) by CPAP can improve oxygenation by increasing functional residual capacity, reducing work of breathing, and recruiting nonaerated alveoli in dependent pulmonary regions [2,3].

As shown by a recent proof of concept study, the extent of changes in oxygenation observed with CPAP in COVID-19 pneumonia is highly variable [4], and the identification of physiological variables reflecting the actual recruitment remains arduous. Moreover, currently, it is not known if the improvement of oxygenation with CPAP may lead to different clinical outcomes in patients with COVID-19 pneumonia. We aimed at investigating if changes in commonly used gas exchange indexes may predict the clinical response to CPAP positioning in patients with COVID-19 pneumonia.

## 2. Materials and Methods

We conducted a multicenter, retrospective study on patients with COVID-19 pneumonia treated with a helmet CPAP (hCPAP) in the High Dependency Respiratory Unit (HDRU) of L. Sacco University Hospital, San Paolo University Hospital and Ospedale Maggiore Policlinico in Milan, Italy, from March 2020 to March 2021. Inclusion criteria were: (i) laboratory-confirmed SARS-CoV-2 infection; (ii) radiological evidence of new-onset pulmonary infiltrates; (iii) a PaO_2_/FiO_2_ ≤ 250 mmHg while on oxygen masks with FiO_2_ ≥ 50% before starting CPAP; (iv) initiation of CPAP within 24 h after HDRU admission.

Assessment of vitals and arterial blood gas analysis (ABG) was performed within 1 h before and 1 h after starting hCPAP. Biochemistry obtained the same day, as well as hospital length of stay and all-cause in-hospital mortality were registered. An hCPAP was started by the protocol as previously reported [5] and was delivered through high-flow generators (VitalSigns Inc., Totowa, NJ, USA; 90–140 L·min^−1^; MYO 3133A, Pulmodyne, Indianapolis, IN, USA) by means of a helmet interface (StarMed, Intersurgical S.p.A., Mirandola, Italy), with an adjustable PEEP valve (VitalSigns Inc., Totowa, NJ, USA) [3,5]. CPAP was administered in 4- or 5-h cycles during the morning and afternoon, as well as during night-time. In critical patients who were not able to tolerate the suspension of CPAP, CPAP was administered continuously and enteral nutrition was thus implemented. Adherence to CPAP and occurrence of possible complications (e.g., damage to the helmet or circuit, lymphedema in the arms, or vomiting) were monitored and managed by the nursing and medical staff. Patients with poor adherence to CPAP, or in whom CPAP had to be prematurely suspended due to intolerance, were excluded from this study.

Ceiling treatment and a “do not intubate” order (DNI) were judged by the treating physician and the critical care physician according to standard local operating procedures and the patients’ will, as previously reported [5]. Indications for endotracheal intubation (ETI) were the presence of respiratory distress, hemodynamic instability, respiratory arrest, a reduction of alertness, or unsatisfactory oxygenation (SpO_2_ < 90% or PaO_2_ < 60 mmHg) while on hCPAP and an FiO_2_ > 60%. Differences in PaO_2_ (ΔPaO2), alveolar-arterial (A-a) O_2_ gradient (ΔA-a O_2_), PaO_2_/FiO_2_ (ΔPaO_2_/FiO_2_), and a modified ROX (mROX) index (partial pressure of oxygen (PaO_2_) to FiO_2_ ratio/RR) (ΔmROX) [6] before and after CPAP positioning were investigated as markers of improved oxygenation during CPAP, also possibly reflecting increased alveolar recruitment.

CPAP failure was defined as either death in the HDRU or the need for ETI and subsequent transfer to the intensive care unit [7].

Analyses were performed with IBM SPSS Statistics for Windows, V.23.0 (Armonk, NY, USA). Variables were expressed as the median and interquartile range (IQR) or means and standard deviation (SD), according to their distribution assessed with the Shapiro-Wilk test. *t*-tests for independent groups, Chi-square, or Mann-Whitney tests were used to compare CPAP successes and failures, as appropriate. A Cox regression analysis was performed to investigate the relationship between changes in PaO_2_, (A-a) O_2_ gradient, PaO_2_/FiO_2_ ratio, and mROX before and during CPAP, and the risk of CPAP failure; the regression was adjusted for either the severity of pneumonia assessed before starting CPAP, by means of CURB-65 or comorbidities by means of the Charlson Comorbidity Index.

The area under the receiver operating characteristic curve (AUROC) was calculated for ΔPaO_2_, ΔA-a O_2_, ΔPaO_2_/FiO_2,_ and ΔmROX, considering improvement thresholds of 20%, 40%, and 60% from baseline. Tests were two-sided and statistical significance was taken at *p* < 0.05. Missing data were not computed for statistical analysis.

The study was conducted in accordance with the amended Declaration of Helsinki (2013) and the study was approved by the Ethical Committees of the involved centers (17263/2020). All patients signed informed consent.

## 3. Results

Two hundred and eleven patients (mean age 64 years, 74% males) were included. Clinical characteristics are reported in Table 1. At baseline, the median PaO_2_ was 68 (57*–*83) mmHg, PaO_2_/FiO_2_ was 129 (91*–*179) mmHg, (A-a) O_2_ was 310 (177*–*559) mmHg, and mROX was 4.77 (3.04*–*7.06). On average, after starting CPAP, PaO_2_, PaO_2_/FiO_2_, and mROX increased by 30 (11*–*68; *p* < 0.001), 57 (12*–*113; *p* < 0.001) mmHg, and 2.5 (0.8*–*5.3) units, respectively, while (A-a) O_2_ was reduced by 68 (−25*–*250; *p* < 0.001) mmHg and the respiratory rate decreased by 2 (−7.5*–*2.0; *p* < 0.001) bpm.

Ninety-three (44%) patients failed hCPAP, of which 42 (19.9%) died in the HDRU, and 51 (24.2%) required intubation. Patients who failed treatment were older, had a higher Charlson comorbidity index and had worse respiratory parameters both before and after starting hCPAP (Table 1). Among the patients who failed CPAP, three developed pneumomediastinum, and one developed pneumothorax (Table 1).

After starting CPAP, the majority of patients showed improvements in all gas exchange indexes, but changes were similar in both successes and failures (Table 1, Figure 1).

Due to the lack of statistical difference in ΔPaO_2_, ΔPaO_2_/FiO_2_, and Δ (A-a) O_2_ gradient between successes and failures, and also due to the high intragroup dispersion of these parameters, only ΔmROX was included in the Cox regression analysis. No significant correlation was found between improvements in mROX and reduced risk of CPAP failure (Table 2).

AUROC for absolute changes in PaO_2_, PaO_2_/FiO_2_, (A-a) O_2_, and mROX to predict CPAP success or CPAP failure for any considered improvement threshold were not statistically significant (Figure 2). When considered as percentage changes from the baseline, AUROC for PaO_2_, PaO_2_/FiO_2_, (A-a) O_2_, and mROX to predict CPAP success or CPAP failure were similarly not significant (data not shown).

## 4. Discussion

Despite the proven effectiveness of CPAP in treating acute respiratory failure due to COVID-19 [8,9], we found high rates of CPAP failure (either death or need for endotracheal intubation) in our cohort. A possible explanation is the high clinical severity of these patients, who, at baseline, showed very compromised oxygenation and elevated inflammation markers, as well as often having significant comorbidities. Moreover, many of these patients were hospitalized in the early stages of the COVID-19 pandemic before the publication of international studies that improved our understanding of therapeutic strategies in this condition, such as the effectiveness of systemic corticosteroids [10] or the ineffectiveness of azithromycin [11] and hydroxychloroquine [12] in reducing mortality. In fact, it has already been observed that patients hospitalized during the 2020 autumn wave in Italy showed lower mortality rates than those hospitalized during the first SARS-CoV-2 pandemic wave, despite similar rates of intubation [13].

We found that patients who failed CPAP were older, had higher baseline IL-6, D-dimer, CURB-65, and blood urea, as well as worse oxygenation parameters and higher respiratory rate, and were more likely to be affected by ischemic heart disease, COPD, and chronic kidney failure. All these findings are coherent with well-known predictors of unfavorable outcomes in COVID-19 pneumonia [7,14,15,16,17,18,19,20,21].

CPAP may increase oxygenation in COVID-19 pneumonia both by improving the ventilation/perfusion ratio and by reducing the pulmonary shunt, recruiting poorly ventilated lung regions [3]. It has been observed that estimation of lung recruitment by PEEP via the recruitment-to-inflation (R/I) ratio can predict clinical outcomes in the invasive setting [22], but this cannot be applied in patients treated with noninvasive ventilation and CPAP. Moreover, there is high interindividual variability in lung recruitment and a subsequent improvement of oxygenation with CPAP [4,23], which is likely due to the heterogeneity in mechanical characteristics and damage of the lungs in these patients [24,25,26,27].

Since increased lung recruitment can be reflected by an increase in the ventilation/perfusion ratio [27] and a reduction of the alveolar-arterial oxygen gradient [15], it would be expected that the observed variations of these parameters after starting CPAP could help estimate the amount of lung recruitment. This consideration would be valuable for clinicians to assess the response to CPAP, and thus to consider the possible need to modulate PEEP or to proceed to intubation and invasive ventilation, since it is known that delaying intubation may lead to poor prognosis in these patients [28,29]. However, it is already known that CPAP can also decrease the work of breathing and, in some instances, the respiratory rate in patients with increased respiratory efforts, thus proving additionally valuable in patients with COVID-19 pneumonia [2,30,31,32]. Therefore, along with PaO_2_, PaO_2_/FiO_2_ ratio, and (A-a) O_2_ gradient, we decided to include mROX as a further marker of response to CPAP. In fact, mROX has been proposed as a superior predictor of clinical outcomes compared to the ROX index in patients with acute respiratory failure treated with HFNC [7] or CPAP [33] and reflects the possible reduction of respiratory rate induced by CPAP in patients with respiratory distress.

However, contrary to expectations, we found that short-term improvements in these parameters poorly correlated with clinical outcomes, and thus we were not able to predict a positive or negative response to the application of hCPAP.

Among the parameters we considered as possible markers of lung recruitment after CPAP, we found no significant difference in PaO_2_, PaO_2_/FiO_2_ ratio, and (A-a) O_2_ gradient between CPAP successes and failures.

Additionally, we found that the absolute increase of mROX, but not its relative increase from the baseline, was greater in successes than in failures; however, punctual increases of mROX were not significantly associated with better outcomes in the respective ROC curve. Moreover, Cox regression showed that improvements in mROX could not predict reduced odds of CPAP failure (Table 2). This contrasts a previous finding that the ROX index may predict better outcomes in severe COVID-19 patients treated with HFNC [34].

There are some possible explanations for this finding. As highlighted by Tobin and collaborators [35], even though the PaO_2_/FiO_2_ ratio is widely employed to assess the severity of hypoxemia in pneumonia and ARDS, the accurate measurement of FiO_2_ could be unfeasible and possibly misleading in a noninvasive setting. The estimation of FiO_2_ may be further complicated in COVID-19 patients treated with a helmet CPAP, due to the use of filters to prevent the dispersion of viral particles [36] and the need to achieve high oxygen flows in order to provide the target FiO_2_ [37]. Moreover, the PaO_2_/FiO_2_ relationship is not linear, but varies with the degree of ventilation–perfusion inequality and shunt, undermining the reliability of comparing PaO_2_/FiO_2_ before and during CPAP [38]. Despite the (A-a) O_2_ being possibly more accurate than the PaO_2_/FiO_2_ ratio in monitoring clinical response to ventilation in CPAP [39], it is still undermined by imprecise measures of FiO_2_.

Second, CPAP grants respiratory support not only by improving oxygenation but also by improving lung mechanics and reducing the work of breathing [40,41]. Individual degrees of this positive effect would not be reflected by short-term modifications of the gas exchange parameters.

Third, clinical deterioration and mortality in severe COVID-19 pneumonia may be also due to acute complications (e.g., acute pulmonary embolism, kidney failure, or cardiovascular involvement) [1,42], which can’t be predicted by gas exchange parameters after starting CPAP.

Finally, the low odds ratio between improvements in oxygenation and the favorable outcome we found in the logistic regression may be attributable to the limited size of the studied cohort.

The available data suggest that CPAP can reduce the need for intubation in patients with COVID-19 pneumonia compared to other forms of respiratory support, such as high-flow nasal cannula [43]. However, delaying intubation may also increase mortality [44]. Therefore, it would be crucial to promptly identify nonresponders to CPAP, who may need evaluation for undergoing timely intubation to prevent further clinical deterioration. Unfortunately, we found that short-term modifications of respiratory parameters were not helpful for this purpose.

This study has limitations. First, the retrospective nature is a possible source of bias. In fact, some clinical information was not available for all patients, such as the time intercurred between the onset of symptoms and the initiation of CPAP treatment. This could have helped the investigation if the response to the application of PEEP might have been affected by the stage of the disease. Moreover, a chest CT scan performed before and after CPAP positioning would have helped identify the mechanisms underlying the improvement of oxygenation during CPAP, as well as clarify the correlation with alveolar recruitment. Third, the effects of the pharmacological interventions, especially regarding systemic steroids and tocilizumab, have not been taken into account as a confounding factor; however, considering the mechanistic rationale of the study and the acute observation of CPAP effects on gas exchange, we believe that the effect of drug therapy may not have affected by a large extent the response to CPAP. On the contrary, the use of systemic corticosteroids and heparin might have influenced mortality and disease progression [45]. In fact, especially during the first month of the pandemic wave, the usage of systemic steroids was low, which might have influenced the clinical outcome [13]. Lastly, prone positioning may increase lung recruitment and oxygenation and decrease the rate of intubation in COVID-19 pneumonia [46,47,48]. Unfortunately, in our cohort, awake-prone positioning was rarely and heterogeneously applied, which has prevented the evaluation of its possible favorable effects.

## 5. Conclusions

In conclusion, this study suggests that improvements in oxygenation after starting CPAP can’t be used to inform the clinical outcomes of the individual patient with severe COVID-19 pneumonia. Further prospective studies are necessary to support these observations and help clarify the underlying pathophysiological mechanisms related to the improvement of oxygenation in COVID-19 patients treated with CPAP.

## Figures and Tables

**Figure 1 jcm-11-07186-f001:**
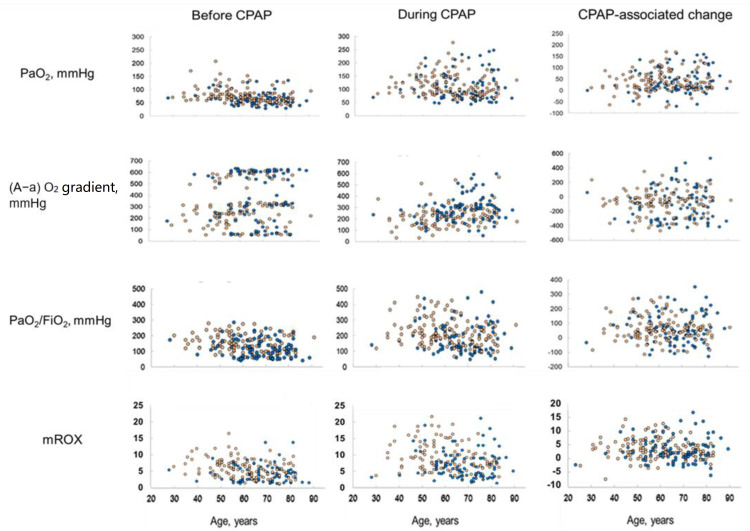
PaO_2_, (A-a) O_2_ gradient, PaO_2_/FiO_2_ ratio, and mROX before and during CPAP, and relative differences. Physiological variables are plotted as a function of age to show their dispersed distribution in different patients. Orange circles = CPAP successes; Blue circles = CPAP failures; PaO_2_ = arterial partial pressure of oxygen; CPAP = continuous positive airway pressure; mROX = PaO_2_/FiO_2_/respiratory rate; (A-a) O_2_ gradient = alveolar-arterial oxygen gradient; FiO_2_ = fraction of inspired oxygen.

**Figure 2 jcm-11-07186-f002:**
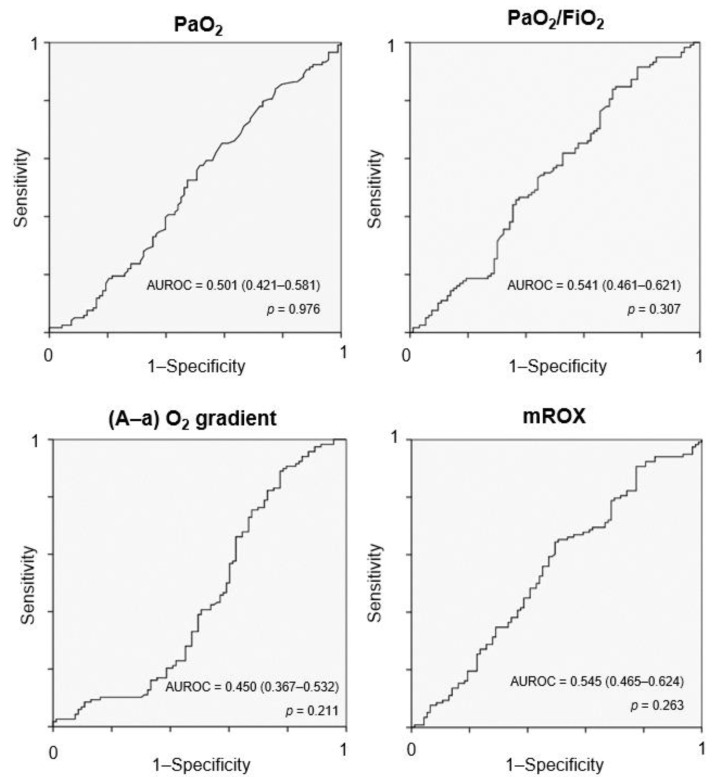
ROC curves for absolute increases in PaO_2_, (A-a) O_2_ gradient, PaO_2_/FiO_2_ ratio, and mROX before and during CPAP, and relative differences. PaO_2_ = arterial partial pressure of oxygen; FiO_2_ = fraction of inspired oxygen; (A-a) O_2_ gradient = alveolar-arterial oxygen gradient; mROX = PaO_2_/FiO_2_/respiratory rate; AUROC = area under the ROC curve.

**Table 1 jcm-11-07186-t001:** Clinical and gas exchange characteristics before and after CPAP positioning in the whole study population and in patients that succeeded or failed respiratory support with CPAP. Values are reported as frequencies or median (IQR), as appropriate. A calculated comparing patients that succeeded and failed CPAP. * *p* < 0.001 for the difference between baseline value and post CPAP.

Characteristics		Entire Study Population (*n* = 211)	CPAP Success (a) (*n* = 118)	CPAP Failure (b) (*n* = 93)	*p*-Value (a vs. b)
Age, years		64 (55–74)	61 (51–72)	67 (60–76)	<0.001
Males, *n* (%)		155 (73.5)	88 (74.6)	67 (72)	0.398
Past medical history					
Charlson Comorbidity Index, score		3 (1–4)	2 (1–4)	2 (3–5)	<0.001
Hypertension, *n* (%)		94 (44.5)	53 (44.9)	41 (44.1)	0.508
Diabetes mellitus, *n* (%)		48 (22.7)	23 (19.5)	25 (26.9)	0.134
Ischemic heart disease, *n* (%)		36 (17.1)	12 (10.2)	24 (25.8)	0.002
Arrhythmia, *n* (%)		17 (8.5)	8 (7.2)	9 (10.1)	0.315
Smoke	Active, *n* (%)	7 (3.4)	5 (4.4)	2 (2.2)	0.613
	Ex, *n* (%)	34 (16.6)	17 (14.9)	17 (18.7)	
	Never, *n* (%)	164 (80)	92 (80.7)	72 (79.1)	
COPD, *n* (%)		15 (7.1)	4 (3.4)	11 (11.8)	0.018
Asthma, *n* (%)		13 (6.2)	8 (6.8)	5 (5.4)	0.452
CKD, *n* (%)		14 (6.6)	3 (2.5)	11 (11.8)	0.008
Immune depression, *n* (%)		11 (5.5)	6 (5.4)	5 (5.6)	0.593
Clinical variables before starting CPAP					
Leucocytes, ×10^3^/L		8.84 (2.07–55)	7.28 (5.60–10.31)	9 (5.81–11.28)	0.168
Urea, mg/dL		38 (28–55)	37 (26–50)	45.5 (30–65.5)	0.004
IL 6, pg/mL		56 (22–134)	45 (14–76)	114.5 (45.75–254.75)	<0.001
CRP, mg/L		24.08 (11.63–93.9)	25 (11.23–81.5)	21.82 (12.45–120)	0.457
Platelets, ×10^3^/L		225.5 (169–302.75)	242 (175–318.5)	207 (144–284)	0.062
D-Dimer, µg/L FEU		929.5 (593–1651.25)	803 (568–1281)	1105 (668–2491)	0.006
CURB 65		2 (1–2)	1 (0–2)	2 (1–3)	<0.001
FiO_2_, %		60 (40–100)	60 (40–60)	60 (40–100)	0.027
pH		7.48 (7.45–7.51)	7.48 (7.45–7.5)	7.48 (7.46–7.51)	0.308
PaO_2_, mmHg		68 (57–83)	72.5 (60–87)	60 (49.5–72.5)	<0.001
PaCO_2_, mmHg		33 (30–37)	34 (30–37.2)	32 (28–35)	0.006
Respiratory rate, bpm		28 (24–32)	27 (22–30)	30 (25–36)	<0.001
PaO_2_/FiO_2_, mmHg		128 (91–179)	141.8 (109.6–194.6)	118.23 (61–165.5)	<0.001
(A-a) O_2_, mmHg		310 (177–559)	280.9 (176.7–333.7)	337.8 (177.95–608.5)	0.001
mROX		4.77 (3.04–7.06)	5.7 (4–7.6)	3.7 (2–5.7)	<0.001
Clinical variables after starting CPAP					
FiO_2_, %		60 (50–60)	50 (50–60)	60 (50–65)	<0.001
PEEP, cmH_2_O		10 (7.5–10)	10 (7.5–10)	10 (7.5–12)	0.088
pH		7.47 (7.44–7.49)	7.46 (7.44–7.49)	7.47 (7.43–7.49)	0.950
PaO_2_, mmHg		100 (79–141)	107.5 (86–145.5)	85 (68–134.5)	0.001
PaCO_2_, mmHg		36 (32–39)	36 (34–40)	34 (31–39)	0.026
PaO_2_/FiO_2_, mmHg		195 (131.7–256.7)	207.5 (165.4–265.8)	145 (110–242.7)	<0.001
(A-a) O_2_, mmHg		240 (188–308)	222 (160.8–274.2)	285.3 (212.8–326.8)	<0.001
Respiratory rate, bpm		24 (22–28)	24 (21–28)	26 (23–30)	0.007
mROX		7.4 (5.1–10.7)	8.3 (6.5–11.3)	6.0 (3.7–9.3)	<0.001
Differences compared with baseline					
ΔPaO_2_, mmHg		30 (11–68) *	32 (11–62)	29 (8.5–75)	0.976
ΔPaO_2_, % increase from baseline		48.3 (13.8–104.8)	46.3 (13.4–90)	50.7 (14.6–128.4)	0.407
ΔPaO_2_/FiO_2_, mmHg		57 (12–113.3) *	60.7 (22–108.6)	51.1 (−4.9–142.7)	0.307
ΔPaO_2_/FiO_2_, % increase from baseline		50.4 (9.1–104.9)	43.4 (13.4–89.6)	66.2 (−4.1–136.6)	0.393
Δ(A-a) O_2_, mmHg		68 (−25–250) *	55 (−7–136)	91 (−68–304)	0.211
Δ(A-a) O_2_, % decrease from baseline		20.7 (−10.6–49.5)	18.2 (−4.4–48.46)	27 (−38.5–51)	0.742
ΔRespiratory rate, bpm		−2 (−7.5–2.0) *	−2.0 (−6.0–3.7)	−3.0 (−10.0–2.0)	0.167
ΔRespiratory rate, % decrease from baseline		8.2 (−10–24.3)	6.7 (−12.2–21.4)	10 (−6.3–27.3)	0.241
ΔmROX		2.5 (0.8–5.3) *	2.7 (1.0–5.4)	2.1 (0.5–5.3)	<0.001
ΔmROX, % increase from baseline		59.1 (13.8–133.6)	53.5 (15.3–111.1)	83 (10–210)	0.212
Clinical outcomes					
CPAP duration, days		6.0 (3.5–10)	6.5 (5.0–12.0)	4.0 (3.0–7.2)	<0.001
Hospital stay, days		15.0 (10–23)	16 (13–24)	8.0 (5.0–20)	<0.001
Pneumomediastinum, *n* (%)		3 (1.4)	0 (0)	3 (3.2)	0.084
Pneumothorax, *n* (%)		1 (0.5)	0 (0)	1 (1.1)	0.441
Intubated, *n* (%)		51 (24.2)	-	51 (54.8)	n/a
Died in HDRU, *n* (%)		42 (19.9)	-	42 (45.2)	n/a

Variables were expressed as the median and interquartile range (IQR) or means and standard deviation (SD), according to their distribution assessed with the Shapiro-Wilk test. CPAP = continuous positive airway pressure; COPD = chronic obstructive pulmonary disease; CKD = chronic kidney disease; IL-6 = interleukin 6; CRP = C reactive protein (upper limit of normal 10 mg/L); FEU = fibrinogen equivalent units; CURB-65 = Confusion, blood Urea nitrogen, Respiratory rate, Blood pressure, and age ≥ 65 score; PaO_2_ = arterial partial pressure of oxygen; PaCO_2_ = arterial partial pressure of carbon dioxide; FiO_2_ = fraction of inspired oxygen; A-a O_2_ gradient = alveolar-arterial oxygen gradient; mROX = PaO_2_/FiO_2_/respiratory rate; HDRU = high dependency respiratory unit.

**Table 2 jcm-11-07186-t002:** Association between changes in mROX before and during CPAP, and risk of CPAP failure. CPAP = continuous positive airway pressure; mROX = PaO_2_/FiO_2_/respiratory rate.

Cox Regression between ΔmROX and CPAP Failure	Adjusted OR	95% C.I.	*p*-Value
Not adjusted	0.963	0.904–1.026	0.239
Adjusted for CURB-65	0.974	0.908–1.046	0.474
Adjusted for Charlson Comorbidity Index	0.989	0.925–1.058	0.755

## Data Availability

The data that support the findings are available from the corresponding author upon reasonable request.

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
