# Peer review of "Acute Improvements of Oxygenation with Cpap and Clinical Outcomes in Severe COVID-19 Pneumonia: A Multicenter, Retrospective Study"

_jcm, 2022, doi:10.3390/jcm11237186_

Round 1

Reviewer 1 Report

A significative bias is that the study is retrospective and covers a period during which many conditions have changed; the authors underlie that different therapies were intoduced, but is not the only variation occurred: in the period between march 2020 and March 2021 almost three epidemic waves occurred. During the pandemic peaks occurred in Lombardy-Italy(in March-April 2020, November 2020, February 2021) the huge number of patients overwhelmed hospital resources, in particular the availability of intensive care beds.

I’m not sure that during March 2020 criteria and timing for intubation and/or DNI orders were applied on the fields as in March 2021: i.e. during November 2020 in Milan was opened the Fiera Hospital, a dedicated hub for Covid intubated patients that increased ICU resources, so that delay in intubation was dramatically reduced.

In my opinion the number of patients is too small considering that were involved four of the biggest hospital in Milan: all pateints treated with hCPAP were included? Was there any kind of selection of patients?

In the results data about adverse effects of hCPAP are lacking.

Were there any episode of pneumothorax and/or pneumomnediastinum?

How was distribution of adverse effects between the two groups?

There are not data about the time from onset of symptoms and initiation of hCPAP: we know that Covid-19 pneumonia evolves and the time of initiation of respiratory support may have impact on outcome.

Reviewer 2 Report

The extent of the study is very limited since this was a prognosis study without randomization or another adjustment. It remains unclear if the results can be readily applicable across all groups since the entry status was uneven. In addition, the timespan is relatively large, meaning that there were undoubtedly substantial changes in the clinical approach, further limiting the results. The design is unclear – you refer to the general population, when you should say the entire sample (Table 1). Then, you compared the CPAP of those who survived and those who did not, but there are significant differences even at the beginning (Table 1). You state: “ After starting CPAP, all gas exchange indexes tended to improve more in patients that succeeded compared with patients that failed CPAP (Table 1), although changes substantially overlapped (Figure 1).“ This is a clear case of confounding by indication and does not provide grounds for a decent conclusion. You need to stratify them according to some general clinical status at admittance (ASA, APACHE, or something similar) in order to be able to compare both groups in the same scale. In addition, the comorbidity status is significantly different between the two groups, requiring further adjustment by the logistic or even better Cox regression. Table 2 is a partial logistic regression, but it is flawed. You have to show the entire model with all significant predictors and a mandatory reporting of the CIs. Only then can it be meaningful. Also, it is better to include variables rather than their changes, as it remains difficult to explain the results adequately. Overall, the Results and the Discussion overestimate the real effect, by focusing on the difference between the groups, which are obviously not just due to COVID. Therefore, I do not think this is acceptable in the current form, but I am willing to re-review the manuscript after implementing these suggestions. 

Round 2

Reviewer 1 Report

I appreciated authors' reply.  I haven't other questions.

Reviewer 2 Report

Thank you for these improvements

Round 3

Reviewer 2 Report

I approve the latest version